# Pathogenic *Leptospira* species identified in dogs and cats during neutering in Thailand

Metawee Thongdee[1], Somjit Chaiwattanarungruengpaisan[1], Weena Paungpin[1], Sivapong Sungpradit[2], Sineenard Jiemtaweeboon[3,4], Ekasit Tiyanun[5], Kanin Ruchisereekul[6], Sarin Suwanpakdee[1,3], Janjira Thaipadungpanit[7,8]*

1 The Monitoring and Surveillance Center for Zoonotic Diseases in Wildlife and Exotic Animals, Faculty of Veterinary Science, Mahidol University, Salaya, Nakhon Pathom Province, Thailand, 2 Department of Pre-Clinic and Applied Animal Science, Faculty of Veterinary Science, Mahidol University, Salaya, Nakhon Pathom Province, Thailand, 3 Department of Clinical Sciences and Public Health, Faculty of Veterinary Science, Mahidol University, Salaya, Nakhon Pathom Province, Thailand, 4 Livestock and Wildlife Hospital, Faculty of Veterinary Science, Mahidol University, Sai Yok, Kanchanaburi Province, Thailand, 5 One Health Animal Clinic, Nakhonsawan Campus, Mahidol University, Nakhonsawan, Thailand, 6 3D Pet Hospital, Bangkok, Thailand, 7 Department of Clinical Tropical Medicine, Faculty of Tropical Medicine, Mahidol University, Bangkok, Thailand, 8 Mahidol-Oxford Tropical Medicine Research Unit, Faculty of Tropical Medicine, Mahidol University, Bangkok, Thailand

* janjira.tha@mahidol.ac.th

## Abstract

Pathogenic species of the genus *Leptospira* cause an underdiagnosed zoonosis in humans and animals called leptospirosis. Animal reservoirs often remain asymptomatic yet shed the active spirochete in urine, making the control of leptospirosis transmission to humans more challenging. Asymptomatic leptospirosis in human companions, such as dogs and cats, resulting in unrecognised infections, has been demonstrated in a few countries. Crucially, the current lack of molecular epidemiology data on *Leptospira* among companion animals in Thailand underscores the urgent need to investigate transmission dynamics for effective regional control. We investigated the prevalence of *Leptospira* infection in cats and dogs during neutering in seven provinces across Thailand. The urine samples were screened for *Leptospira* DNA by PCR targeting the *rrs* gene and further speciation using the Sanger Sequencing Analysis. The 56/567 (9.9%) animals were positive for *Leptospira* in the Pathogen clade, including 34/303 (11.2%) dogs and 22/264 (8.3%) cats. The partial *rrs* gene analysis identified *L. interrogans*, *L. weilii*, and *L. borgpetersenii* (4.4%) as well as Pathogen subclade 2 species (1.4%). Notably, this study reports the first molecular detection of *L. yasudae* (1.0%) in companion animals in Thailand. The identification of these three key pathogenic *Leptospira* species, common causes of human leptospirosis in Southeast Asia, in clinically healthy owned and free-roaming dogs and cats, suggests the risk of human leptospirosis in the areas investigated. These companion animals, often living in close contact with human, may contribute to daily risks. Therefore, enhanced surveillance and vaccination programs for dogs and

**Data availability statement:** All relevant data are within the manuscript and its Supporting Information files.

**Funding:** This work was funded by the Mahidol University (Award no. BRF1-A38/2564) to MT. The funders had no role in study design, data collection and analysis, publication decision, or manuscript preparation.

**Competing interests:** The authors have declared that no competing interests exist.

cats, coupled with targeted public awareness campaigns, are critical for mitigating the risk of human infections.

## Author summary

Leptospirosis, a neglected zoonotic disease caused by pathogenic species of genus *Leptospira*, can be transmitted to humans through contact with infected animal body fluids or contaminated environments. Free-ranging dogs and cats can shed infectious *Leptospira* in their urine, posing a significant risk. In Thailand, animal vaccine coverage for diseases like leptospirosis in these animals is unknown and unregulated, raising concerns about zoonotic transmission and reflecting a lack of routine veterinary support. Our study joined non-profit, volunteer-led neutering programs that travel to various provinces, offering free services for owned and unowned pets. This allowed our team to collect urine specimens directly from the urinary bladder for the detection of *Leptospira*. We found nearly 10% of the recruited asymptomatic dogs and cats were infected by the molecular screening assay. Three pathogenic *Leptospira* species, commonly infecting humans globally, were identified. Notably, we report the first detection of *L. yasudae* in animal urine samples, providing evidence of infectivity for a species not previously recognised for its pathogenicity and typically reported from the environment. This highlights the risk of human leptospirosis from contact with cats and dogs, emphasising the need for public awareness and annual vaccination programs for all pets.

## Introduction

The genus *Leptospira* comprises Gram-negative spirochetes. The primary pathogenic species that cause human leptospirosis worldwide are *L. interrogans* and *L. borgpetersenii* [1,2]. Many mammals, including rodents, dogs, cats, pigs, bats, and cattle, serve as reservoir hosts for spirochetes, harbouring the bacteria in their proximal renal tubules without showing clinical signs [3,4]. They excrete living bacteria in their urine, which can transmit the disease through direct or indirect exposure to contaminated environments [3–5]. Disease severity depends on bacterial virulence, inoculum size and host immunity [5].

Of 68 species in the genus *Leptospira*, 40 are categorised into the Pathogen (P) clade, based on whole genome sequence analysis [6–12]. The P clade comprises two subclades (P1 and P2). Of the 19 species in subclade P1, eight (including *L. interrogans*, *L. weilii* and *L. borgpetersenii*) are key species that cause leptospirosis worldwide. The other eleven species (including *L. alstonii, L. kmetyi* and *L. yasudae*) are recently discovered species isolated from environmental samples, with no evidence of human or mammalian infections. Of 21 species of the P2, five (including *L. wolffii* and *L. licerasiae*) have been known as the intermediate group, causing mild leptospirosis in mammals and humans [11,13–17].

Human leptospirosis was commonly reported in tropical and subtropical regions [18,19]. In 2015, Costa et al. estimated over 1 million global human infections and more than 58,000 fatal cases (using data from 1970 to 2008) [18]. In Thailand, 24,378 human patients were reported to the Bureau of Epidemiology, Ministry of Public Health between 2017 and 2021 (average incidence rate of 3.6 per 100,000 population per year) [20]. The incidence rates in the lower northeastern (11.5 per 100,000 population per year) and upper southern (9.7 per 100,000 population per year) regions were higher than those in other regions of Thailand. Computational models showed that leptospirosis transmission is amplified by environmental changes, specifically increased rainfall and flood exposure. Furthermore, using diverse and potentially contaminated household water sources and occupational contact with moist soil, environmental water, or infected animals significantly raise the risk for all hosts (animals and humans) [21–23].

Leptospirosis in animals can pose a public health threat, as asymptomatic infected animals can spread living bacteria through urination, contaminating habitats and foraging areas where humans live, travel, or work. The more exposure to a variety of probable infection sources, the higher the risk of infections, which shows the complexity of disease control [24,25]. Companion animals, especially free-roaming dogs and cats, act as a critical bridge for *Leptospira* transmission. They acquire pathogens from primary reservoirs, such as rodents and contaminated environments, and then introduce them into households, where the disease can be passed to owners through close, intimate exposures. In Thailand, the pet population was estimated at over 12 million between 2019 and 2020, with a high-risk group of approximately one million free-roaming animals [26–29]. The prevalence of *Leptospira* carriage in free-roaming and client-owned animals varies across the country's geographic regions [30–32].

*Leptospira* infections (symptomatic or asymptomatic) have been reported in many countries using PCR or culture, with median rates of 7.3% (IQR = 4.4-15.7%) in dogs and 6.3% (IQR = 3.3-22.1%) in cats since 2003 [30–70]. However, cross-sectional studies of asymptomatic *Leptospira* infections and carriages in dogs and cats in Thailand are scarce [30–32]. A variety of *Leptospira* P1 species are commonly reported globally. For instance, *L. interrogans* and *L. borgpetersenii* have been found in both dogs (0.4-16% vs 0.5-1%) [25,31,33,34,36,37,41,42,47–52,55–57,71] and cats (2–5% vs 0.7-7%) [58,61–64,70], while other species like *L. weilii* (1–3%) [31,33], *L. santarosai* (0.5-37%) [42,50,54,57], *L. noguchii* (2–37%) [25,50], *L. kirschneri* (0.4%) [52] and *L. kmetyi* (3–10%) [33,48] were found predominantly in dogs. In comparison, the P2 (*L. wolffii* and *L. licerasiae*) infections were reported in 18% and 5% of dogs originating from Iran and Sri Lanka, respectively [48,71]. In this study, we aimed to determine the prevalence of *Leptospira* carriage in dogs and cats across three regions of Thailand, establishing current molecular epidemiological data essential for directly informing targeted leptospirosis control strategies.

## Methods

### Ethics statement

The collection and analysis of animal samples were conducted between 2020 and 2023 under the approval of the Institute for Animal Care and Use Committee, Faculty of Veterinary Science, Mahidol University (MUVS-2020-01-03). The national standard's animal welfare guidelines were followed. Written informed consent for participation was obtained from the animals' owners. For unowned or free-roaming dogs and cats collected from public areas (such as temples and schools), informed consent for a sample collection was obtained from the heads of each community. Laboratory protocols were approved by the Institutional Biosafety Committee of the Faculty of Veterinary Science, Mahidol University (IBC/MUVS-B-001/2564).

### Sample size calculation

Sample size estimation was performed using ProMESA software version 2.3 (EpiCentre, Massey University, New Zealand) based on the previously reported leptospirosis prevalence at 10% [31] of dogs and 0.8% of cats [32] at a 95% confidence interval and a 0.2 acceptable relative error. We used the maximum of the generally recommended relative error as

the unknown prevalence in the study regions. This resulted in a minimum sample size of 223 dogs and 236 cats, which was required for the study.

## Study population

Clinically healthy dogs and cats underwent sterilisation through the neutering program of the Faculty of Veterinary Science, Mahidol University and nonprofit organisations. All owned (n = 271) and free-roaming/unowned (n = 296) dogs and cats were included in this study. The neutering program was carried out across seven provinces in central, western, and southern Thailand (Fig 1). The sex and estimated age of each animal were recorded.

## Dog and cat urine collections and DNA preparations

Urine samples were collected during trips with the non-profit, volunteer-led neutering programs. These programs provided access to anesthetised animals from which urine samples were collected using sterilised urinary catheters before

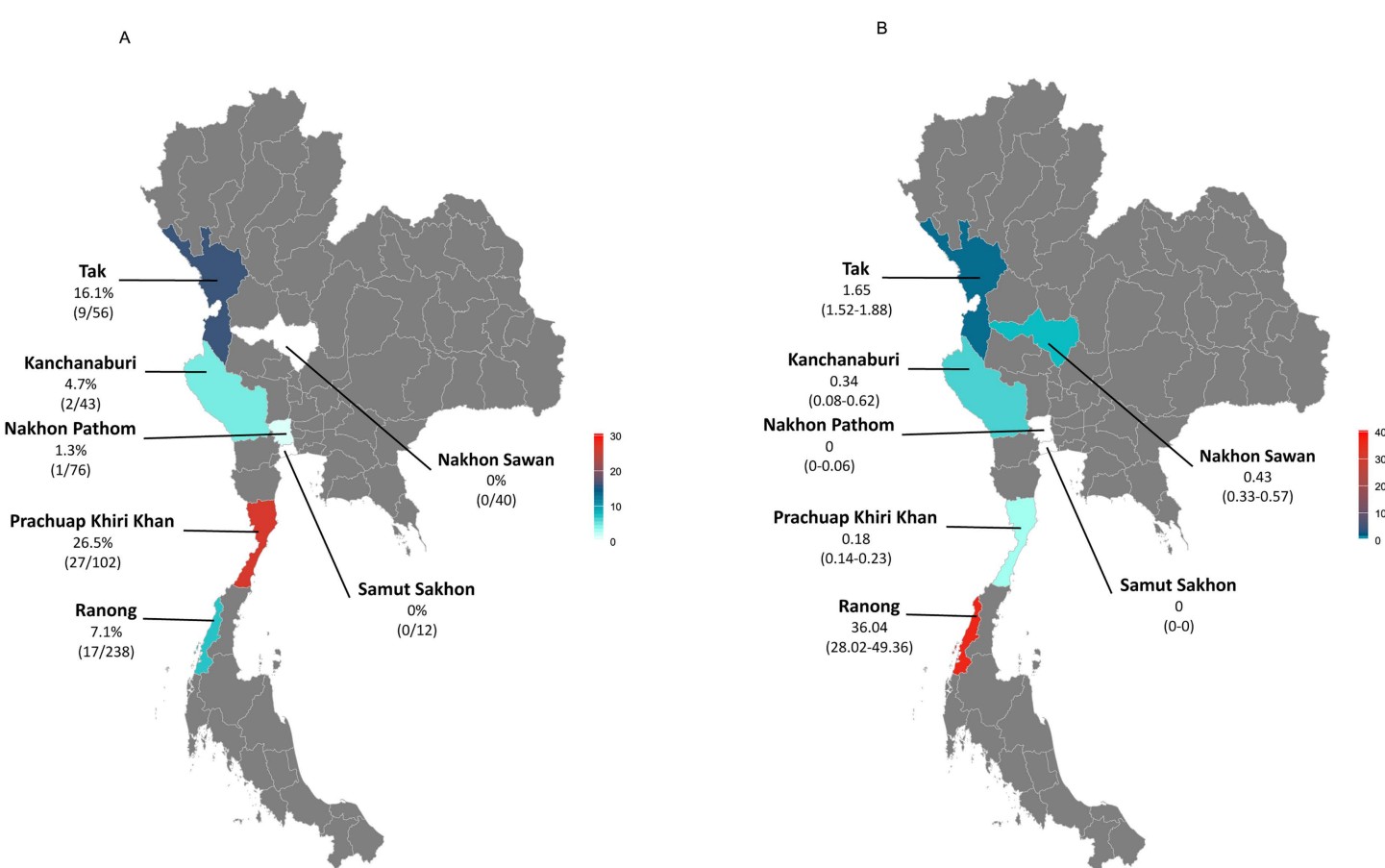

**Fig 1. Geographic distribution of dog and cat *Leptospira* infections (Panel A) and median annual human leptospirosis incidence (Panel B) in seven provinces of Thailand: Nakhon Sawan, Nakhon Pathom, and Samut Sakhon (Central); Tak, Kanchanaburi, and Prachuap Khiri Khan (West); and Ranong (South).** Infection frequencies in animals were identified using PCR and sequencing. Human leptospirosis incidence data (per 100,000 population) were reported by the Ministry of Public Health. Darker shades indicate higher prevalence/incidence. Base map shapefiles were obtained from the Humanitarian Data Exchange (https://data.humdata.org/dataset/cod-ab-tha)(available under the Creative Commons Attribution for Intergovernmental Organisations license--https://data.humdata.org/faqs/licenses).

sterilisation surgery. We collected a single urine sample from each animal. Available urine volume in the urine blader was collected into a new sterile 50 ml tube and kept on ice upon collection. The urine samples were transported to laboratories at cool temperatures (2°C to 6°C) and then stored at -80°C until used for DNA preparation. Collected urine specimens' volume ranged from 1-370 ml (median = 42 ml) in dogs and 0.8–108 ml (median = 9.5 ml) in cats, which depended on each animal's size. The total volume of each collected sample contained in 50 ml tubes was centrifuged at 2,000 relative centrifugal force (RCF) for 10 minutes at 4°C to remove cells and debris. Then, the supernatant was transferred to a new sterile 50 ml tube and centrifuged again at 20,000 RCF for 30 minutes at 4°C to obtain a microorganism cell pellet for the DNA preparation using the Genomic DNA Mini Kit (blood and cultured cells) (Geneaid, New Taipei City, Taiwan) according to the manufacturer's protocol. The extracted DNA samples were eluted using 30–40 µl of the elution buffer.

### Identification of *Leptospira* species based on partial 16S rRNA sequences amplification and analysis

The nested PCR assay targeting the partial 16S rRNA sequences of the P1 and P2 was conducted as described previously [72], with modifications as described in S1 File. The expected 547-bp amplicons were visualised using 1.5% agarose gel electrophoresis. Then, the PCR products were purified from agarose gel using GenepHlow Gel/PCR Kit (Geneaid, New Taipei City, Taiwan). It was conducted as described in the manufacturer's protocol. The purified products were sent to Bionics (South Korea) for the Sanger DNA Sequencing.

The chromatogram results were inspected and edited for consensus using BioEdit Sequence Alignment Editor version 7.0.5.3. Maximum likelihood trees were reconstructed from the trimmed 443-nucleotide partial 16S rRNA gene alignments (from position 63–505 based on *L. alexanderi* GenBank accession number: NR_043047.1) using MEGA software version 11 [73]. The details of the algorithm used for the analysis were described in the S1 File. All sequences from this study (accession numbers: OQ446624-OQ446662)(S1 Table) and the 57 reference sequences of 40 *Leptospira* spp. acquired from GenBank data used for the analysis, (S2 Table). The phylogenetic tree was displayed and annotated using the Interactive Tree Of Life version 6.0 [74].

### Statistical analysis

Data management was performed using Microsoft Excel, and statistical analyses were conducted using R version 4.0.3 [75]. The prevalence of *Leptospira* spp. infection was stratified by sex, age group, ownership, and study site. To determine the strength of associations between prevalence and exposure factors, Odds Ratios (ORs) and 95% confidence intervals (CIs) were calculated using the 'epitools' package for R [76]. Small-sample adjustments were applied to both OR estimates and CIs (using a normal approximation), and p-values were derived using Fisher's exact test. Statistical significance was defined as a $p$-value < 0.05 and a 95% CI that did not include 1. Adjustments for multiple comparisons were not applied, as the primary objective was to estimate the magnitude of associations rather than strictly test hypotheses. Finally, the geographical distribution of *Leptospira* prevalence was visualised via choropleth mapping using the ggplot2 and sf packages in R. Base map shapefiles for Thailand's administrative boundaries were obtained from the Humanitarian Data Exchange (URL: https://data.humdata.org/dataset/cod-ab-tha), sourced from the Royal Thai Survey Department. The maps were plotted with *Leptospira* spp. carriage among dogs and cats (data from this study) and human leptospirosis incidence in Thailand (S2 File)

## Results

### Asymptomatic *Leptospira* infections in dogs and cats

A total of 567 animals, comprising 303 dogs (53.4%) and 264 cats (46.6%), clinically healthy, were recruited into the study during the neutering service. They originated from seven provinces in three regions of Thailand, with the infection frequencies shown in Fig 1. The 5/7 provincial sites show infections ranging from 1% to 16% of the recruited animals.

The infection rates are likely underestimated because the study design only detects chronic carriers (leptospiruria) and is subject to the inherent challenge of intermittent bacterial shedding in the urine.

The overall *Leptospira* infection prevalence was 11.2% (34/303) in dogs and 8.3% (22/264) in cats. In both animal types, statistical analysis revealed no significant differences in prevalence by sex, age group, or ownership status (Table 1). Although age was not statistically significant, no infections were detected in juvenile cats, compared with an 8.8% rate in adults. Geographical location was a significant predictor of infection for dogs. Dogs in Tak and Prachuap Khiri Khan exhibited significantly higher infection risks compared to the Nakhon Pathom site ($p < 0.05$).

National annual incidence data reported in those seven provinces (S1 File) indicate that Ranong had the highest human leptospirosis incidence (median 36.04 cases per 100,000 population), nearly 22-fold higher than Tak (1.65/100,000)(Fig 1). In contrast, our study found the highest leptospirosis prevalence in dogs and cats in Prachuap Khiri Khan (26.5%) and Tak (16.1%).

### *Leptospira* pathogen clade detected in dogs and cats

The ML tree analysis revealed that 39/56 PCR-positive dogs and cats were infected with *Leptospira* subclade P1 and P2 (S1 File). Five distinct clusters were identified (Fig 2): Interrogans (46%), Borgpetersenii (13%), Weilii (5%), Yasudae (15%) and undetermined species of P2 (21%). The P2 classifications, based on *rrs* sequence

**Table 1. Distributions of leptospirosis in dogs and cats stratified by study sites, sex, age group and ownership.**

| Demography | Dogs | | | | Cats | | | |
|---|---|---|---|---|---|---|---|---|
| | n infected (%) | Total (n) | Odds ratio (95%CI) | *P*- value | n infected (%) | Total (n) | Odds ratio (95%CI) | *P*-value |
| **Animals[a] (n=567)** | 34 (11.2) | 303 | 1.325 (0.789-2.412) | 0.2623 | 22 (8.3) | 264 | ref | |
| **Study site[a]** | | | | | | | | |
| Nakhon Sawan (n=40) | 0 | 20 | n/a | n/a | 0 | 20 | n/a | n/a |
| Samut Sakhon (n=12) | 0 | 0 | n/a | n/a | 0 | 12 | n/a | n/a |
| Nakhon Pathom (n=76) | 1 (1.4) | 74 | ref | | 0 | 2 | n/a | n/a |
| Kanchanaburi (n=43) | 1 (4.5) | 22 | 1.659 (0.338-34.566) | 0.408 | 1 (4.8) | 21 | ref | |
| Tak (n=56) | 5 (20.8) | 24 | **9.125 (2.123-89.954)\*#** | **0.003\*** | 4 (12.5) | 32 | 1.379 (0.312-14.913) | 0.637 |
| Prachuap Khiri   Khan (n=102) | 18 (32.7) | 55 | **17.290 (4.373-133.623)\*#** | **<0.001\*** | 9 (19.1) | 47 | 2.308 (0.556-20.472) | 0.157 |
| Ranong (n=238) | 9 (8.3) | 108 | 3.285 (0.814-26.874) | 0.050 | 8 (6.2) | 130 | 0.6504 (0.157-5.724) | 1.000 |
| **Sex[b]** | | | | | | | | |
| Male (n=181) | 10 (8.8) | 114 | ref | | 5 (7.5) | 67 | ref | |
| Female (n=353) | 23 (12.8) | 179 | 1.385 (0.693-3.223) | 0.345 | 17 (9.8) | 174 | 1.1118 (0.464-3.439) | 0.803 |
| Missing (n=33) | 1 (10) | 10 | 0.946 (0.310-1.443) | 1.000 | 0 | 23 | n/a | n/a |
| **Age[b]** | | | | | | | | |
| Junior (n=46) | 2 (6.1) | 33 | ref | | 0 | 13 | n/a | n/a |
| Adult (n=521) | 32 (11.9) | 270 | 1.384 (0.450-6.550) | 0.556 | 22 (8.8) | 251 | n/a | n/a |
| **Ownership[b]** | | | | | | | | |
| Owned (n=271) | 11 (10.2) | 108 | ref | | 12 (7.4) | 163 | ref | |
| Free-roaming (n=296) | 23 (11.8) | 195 | 1.075 (0.547-2.440) | 0.709 | 10 | 101 | 1.263 (0.588-3.288) | 0.497 |

[a]Calculate the odds ratio by the small sample adjustment and comparing odds to the least animal Leptospira infection prevalence in dog or cat samples.

[b]The odds ratio (the small sample adjustment) of dog and cat leptospirosis between the two provinces in the study, using the lowest prevalence as a reference. While sex, age and ownership parameters, male, junior and owned were used as references, respectively.

*Notify the statistically significant:p-value less than 0.05, which was calculated using Fisher's exact method, and the entire range of the 95% confidence interval is more than 1 or less than 1, which was calculated using normal approximation with the small sample adjustment.

#The large interval of 95% confidence intervals suggests a limitation due to the small sample size.

PLOS Neglected Tropical Diseases

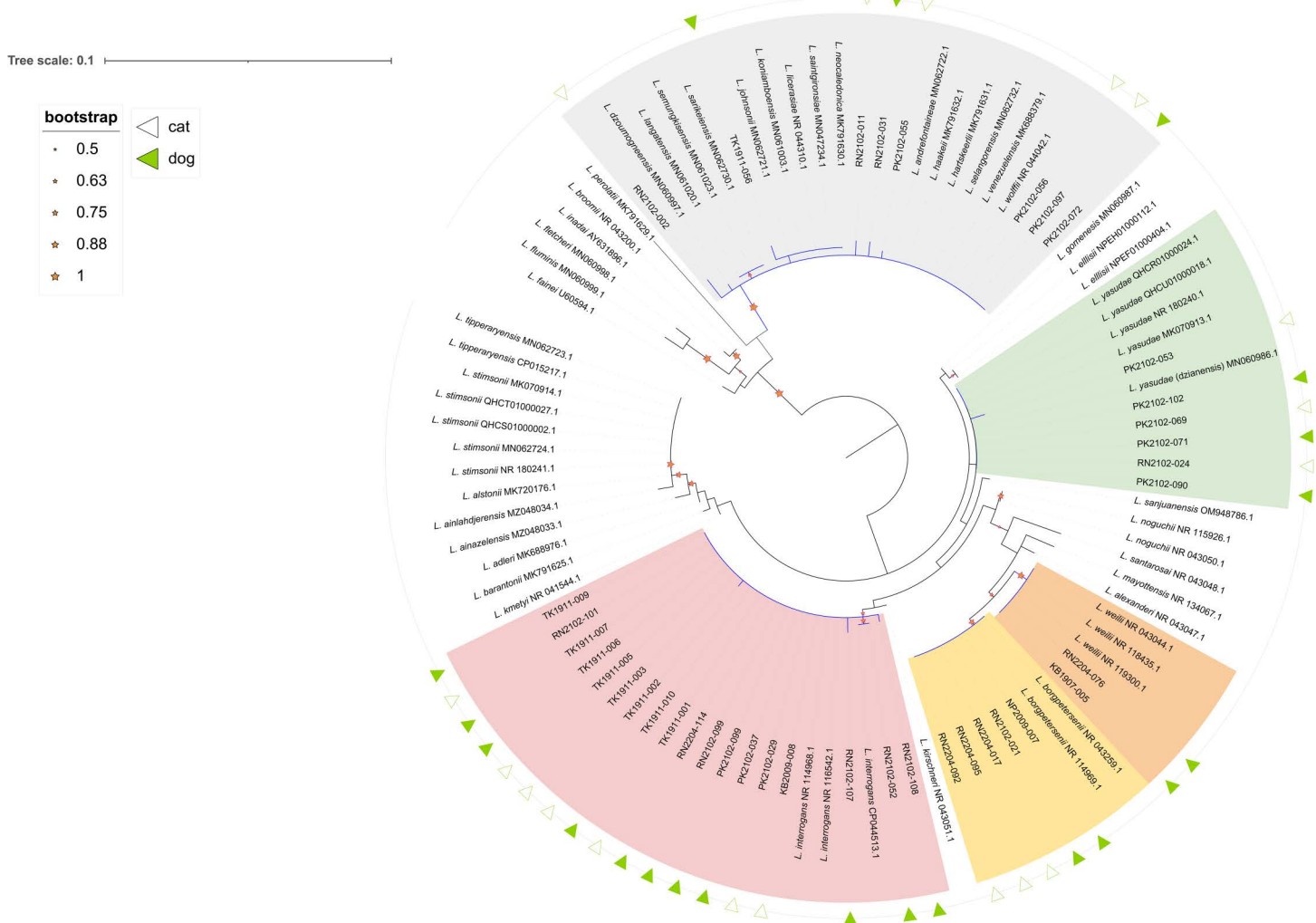

**Fig 2. Maximum Likelihood phylogenetic tree reconstructed using the partial 16S rRNA sequences of 39 nested PCR products amplified from urine samples of dogs (filled green triangles) and cats (not filled triangles).** There were 57 reference sequences of 40 *Leptospira* in the Pathogen clade, which were used to infer the species of amplicon sequences. The *Leptospira* sequences from this study were grouped into five clusters: *L. interrogans* (pink), *L. borgpetersenii* (yellow), *L. weilii* (orange), *L. yasudae* (green) and species in the Subclade 2 (grey) of the Pathogen clade. The sequences grouped in each cluster with a bootstrap value higher than 50% are indicated with a red star.

polymorphisms and the distribution of infection, are detailed in S1 File. The distributions of each subclade are detailed in S3 Table.

Geographically, P1-1 infections (*L. interorgans, L. borgpetersenii* and *L. weilii*) were most prevalent in animals from Tak (16.7% dogs vs 12.5% cats), followed by Ranong (6.5% vs 3.1%), Kanchanaburi (4.6% vs 4.8%), Prachuap Khiri Khan (3.6% vs 2.1%) and Nakhon Pathom (only 1.4% dogs) (Fig 3). Statistical analysis demonstrated that animals in Tak have a significantly higher risk of P1-1 infections than those in Nakhon Pathom (*p*-value=0.005) (Table 2). *L. yasudae* (P1-2) was mainly detected in Prachuap Khiri Khan (5.5% vs 4.3%) and, to a lesser extent, in Ranong (0.8%, cats only). Animals from Prachuap Khiri Khan had a significantly higher risk of *L. yasudae* infections than those from Ranong (*p*-value=0.010). The P2 cluster was most frequently found in Prachuap Khiri Khan (1.8% vs 6.4%), followed by Tak (4.2%, dogs only) and Ranong (0.9% vs 1.5%); the risk of P2 infection did not differ significantly among these three sites.

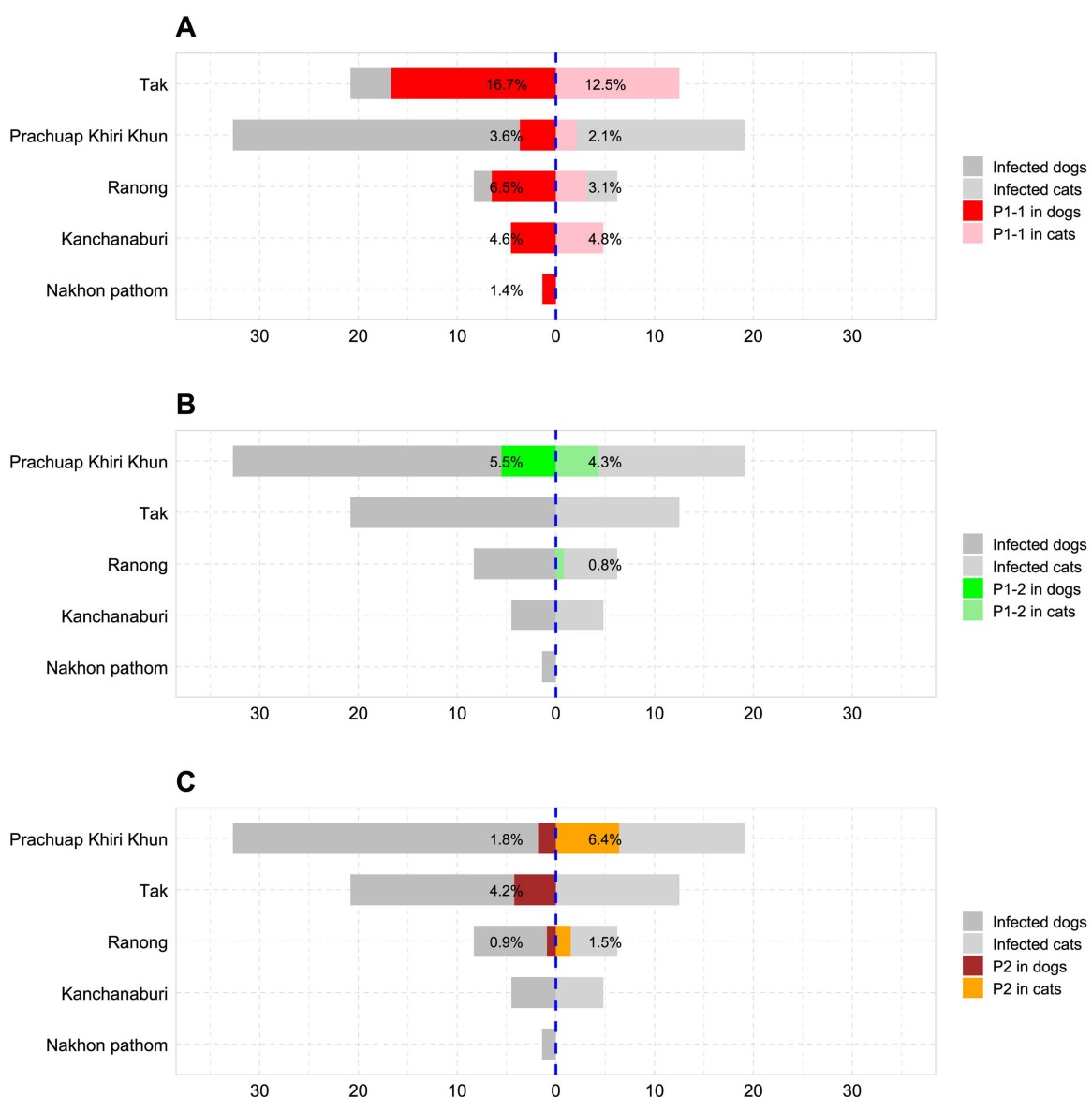

**Fig 3. The funnel graph illustrates the distributions of Pathogen clade (P)- infected dogs (left) and cats (right) in five prevalence study sites, compared to infected animals (PCR-positive without molecular typing).** *L. interrogans*, *L. borgpetersenii* and *L. weilii* (belonging to Group 1 of the Subclade 1 in the P clade as P1-1), *L. yasudae* (belonging to Group 2 of the Subclade 1 as P1-2), and species in the Subclade 2 (P2) were in Panels A, B and C, respectively.

## Discussion

This study reveals a critical public health risk: the shedding of pathogenic *Leptospira* in the urine of clinically healthy (asymptomatic) dogs and cats across central, western, and southern Thailand. Notably, 9.9% of these animals (11.2% dogs vs 8.3% cats) were identified as active shedders, posing a direct transmission risk to owners, other animals, and the environment. Regional analysis indicated that infection rates were lower in the central region compared to the western and southern regions. Given the national reports of human leptospirosis incidence, these findings suggest a potential for disease transmission from dogs and cats to humans within the same spatiotemporal context. However, companion animals

**Table 2. Distribution of *Leptospira* Pathogen clade detected in dogs and cats in each study site.**

| | P1-1 (%) | odds ratio[b] (95%CI) | P value | P1-2 (%) | odds ratio (95%CI) | P value | P2 (%) | odds ratio[b] (95%CI) | P value |
|---|---|---|---|---|---|---|---|---|---|
| **Animal[a]** (n=567) | 25 (4.4) | | | 6 (1.1) | | | 8 (1.4) | | |
| **Study sites[b]** | | | | | | | | | |
| Nakhon Sawan (n=40) | 0 | n/a | | 0 | n/a | | 0 | n/a | |
| Samut Sakhon (n=12) | 0 | n/a | | 0 | n/a | | 0 | n/a | |
| Nakhon Pathom (n=76) | 1 (1.3) | ref | | 0 | n/a | | 0 | n/a | |
| Kanchanaburi (n=43) | 2 (4.7) | 1.786 (0.387-23.775) | 0.296 | 0 | n/a | | 0 | n/a | |
| Tak (n=56) | 8 (14.3) | **6.122 (1.498-51.936)*#** | **0.005*** | 0 | n/a | | 1 (1.8) | 1.049 (0.263-12.591) | 0.573 |
| Prachuap Khiri Khan (n=102) | 3 (2.9) | 1.125 (0.255-12.271) | 0.637 | 5 (4.9) | **6.046 (1.446-55.164)*#** | **0.010*** | 4 (3.9) | 2.374 (0.746-12.673) | 0.204 |
| Ranong (n=238) | 11 (4.6) | 1.809 (0.455-14.227) | 0.306 | 1 (0.4) | ref | | 3 (1.3) | ref | |
| **Gender[b]** | | | | | | | | | |
| Male (n=181) | 4 (2.2) | ref | | 2 (1.1) | ref | | 2 (1.1) | ref | |
| Female (n=353) | 21 (5.9) | 2.232 (0.909-7.160) | 0.054 | 4 (1.1) | 0.682 (0.195-4.384) | 1.000 | 6 (1.7) | 1.029 (0.309-5.839) | 0.723 |
| Missing (n=10) | 0 | n/a | | 0 | n/a | | 0 | n/a | |
| **Age[b]** | | | | | | | | | |
| Junior (n=46) | 0 | n/a | | 1 (2.2) | 1.870 (0.496-19.312) | 0.400 | 1 (2.2) | 1.397 (0.382-13.398) | 0.494 |
| Adult (n=521) | 25 (4.8) | n/a | | 5 (1.0) | ref | | 7 (1.3) | ref | |
| **Ownership[b]** | | | | | | | | | |
| Owned (n=271) | 9 (3.3) | ref | | 2 (0.7) | ref | | 5 (1.8) | ref | |
| Free-roaming (n=296) | 16 (5.4) | 1.492 (0.719-3.675) | 0.306 | 4 (1.4) | 1.224 (0.350-7.852) | 0.688 | 3 (1.0) | 1.372 (0.489-6.678) | 0.489 |

For this table, P1-1 indicated three species: *L. interrogans*, *L. borgpetersenii*, and *L. weillii,* which belong to the Pathogen clade in Group 1 sub-group 1, while P1-2 indicated *L. yasudae,* which belongs to the Pathogen clade in Group 2 sub-group 2. The P2 indicated that the undetermined species belong to Pathogen Group 2.

[a]Calculate the odds ratio by small sample adjustment and comparing odds to the least pathogenic *Leptospira* groups' prevalence in dog or cat samples.

[b]Odd ratio of the pathogenic *Leptospira* group between two provinces in the study, using the lowest prevalence as a reference. While sex, age and ownership parameters, male, junior and owned were used as references, respectively.

*Notify the statistically significant:*p*-value less than 0.05, which was calculated using Fisher's exact method, and the entire range of the 95% confidence interval is more than 1 or less than 1, which was calculated using normal approximation with the small sample adjustment.

#The large interval of 95% confidence intervals suggests a limitation due to the small sample size

are likely not the only sources of human infections in these provinces; other reservoirs (such as rodents and livestock) and environmental exposure via contaminated water or agricultural activities undoubtedly play significant roles in the transmission cycle.

The urinary shedding prevalence in this study was slightly higher than that reported in previous Thai studies [30–32,43]. In dogs, our findings were comparable to those from Malaysia, the USA and several European nations [34,38,41,45,47], but remain lower than the high-burden regions of South Asia, South America and Iran, where the prevalences ranged between 10.6% and 97.4% [33,37,39,42,46,48,50,53,56]. In cats, our prevalence fits within the broad international range of 0.3% to 12.9% reported in several countries of Asia, Europe, Latin America and the Caribbean [49,58,59,61–67,69], contrasting with the strikingly high prevalence reported in Taiwan (67.8%) [60]. Variations in the urinary shedding prevalence may be attributed to geographical location, study population, sample size, season, and screening techniques. The application of molecular methods to specifically identify pathogenic *Leptospira* DNA in clinically healthy carriers provides a crucial tool for monitoring transmission risks from these animals and informing targeted control strategies.

Several *Leptospira* spp. within the Pathogen Clade have been reported in asymptomatic dogs and cats worldwide. Commonly identified pathogenic species include *L. interrogans*, *L. borgpetersenii*, *L. weilii*, *L. kirschneri*, *L. santarosai*, and *L. noguchii* [25,31,33,34,36,37,41,42,47–55,57,58,60–64,68]. Among these, *L. interrogans* predominates in both dogs and cats from Malaysia [34,58] and is a primary species identified in North America & the Caribbean, South America, Europe, Africa Asia and Australia [25,33,36,37,42,47,52,55,58–60,62,63,68]. Consistent with these global trends, this study confirms that dogs and cats across Thailand carry these major pathogenic species. We identified *L. interrogans, L. borgpetersenii, and L. weillii* (which are the predominant causes of human leptospirosis in Thailand and neighbouring countries [77–84]) in dogs and cats during neutering procedures from western (Tak, Kanchanaburi, and Prachuap Khiri Khan), southern (Ranong), and central (Nakhon Pathom) regions.

Notably, beyond these established P1-1 pathogens, we report the first identification of *L. yasudae* (a species of undetermined virulence) in companion animals in western (Prachuap Khiri Khan) and southern (Ranong) Thailand. Furthermore, unspecified species of *Leptospira* within the P2 subclade (associated with mild diseases in humans [17]) was identified in southern (Ranong) and western (Tak and Prachuap Khiri Khan) regions.

We detected *L. interrogans* in similar proportions in urine samples from dogs (3.6%) and cats (2.7%). In contrast, a previous report from northern Thailand (Nan) detected urinary shedding of *L. interrogans* in 6.9% of dogs, but not in cats [31]. Conversely, a study in Songkhla (southern Thailand) found a higher prevalence of *L. interrogans* in cats compared to dogs (0.5% of dogs' vs 7.8% of cats' blood PCR-positive) [43]. Nan and Songkhla were provinces where human leptospirosis was a significant problem, particularly during the wet season [21,85]. The reasons for the variation in host infection rates across these areas remain unclear; further investigation is needed to determine if these host disparities result from local ecological exposure or sampling differences.

*L. borgpetersenii*, a common cause of bovine leptospirosis, was first isolated from US slaughter cattle [86]. However, *L. borgpetersenii* infections have also been identified in dogs and cats. Notably, the number reported among cats was higher than that among dogs. For example, 7.1% of cats were infected on Okinawa Island, Japan [64], and 4.5% in southern Italy [61]. In contrast, 1.1% of dogs in Sri Lanka [33] and 0.5% in Germany [41] were infected. To our knowledge, the identification of *L. borgpetersenii* infections among asymptomatic dogs and cats has been reported for the first time in Thailand: in central (Nakhon Pathom) and southern (Ranong) regions.

*L. weilii* infects a broad host range, including livestock, canines and wildlife [31,87,88]. In this study, *L. weilii* infections were found in dogs with a low prevalence of 0.7% (Kanchanaburi and Ranong) and were not identified in cats. This contrasts with the higher canine infection rates in northern Thailand (3.4%) [31] and Sri Lanka (1.1%) [33].

*L. yasudae* (conspecific with *L. dzianensis* [7]) was previously isolated solely from environmental samples, without documented reports of natural infection in humans or animals. Here, we provide the first description of *L. yasudae* infections in companion animals. Crucially, *L. yasudae* was detected in urine collected aseptically from the bladders, ruling out environmental contamination. These infections were identified predominantly in dogs and cats from Prachuap Khiri Khan and in a cat from Ranong.

Because of the limited diversity in the selected *rrs* sequences, *Leptospira* P2 could not be fully speciate. In stead, these undetermined species were classified into *L. wolffii*- or *L. licerasiae*-related groups (S1 File). Members of P2 have been primarily isolated from environmental samples worldwide, including Mayotte, Puerto Rico, Japan, and Thailand [12,82,89–91]. The potentially pathogenic *L. wolffii* was first documented in a patient from northeastern Thailand and subsequently identified in local rodents [17,92]. Recently, *L. wolffii* infections have been reported in other regions, including Iran and north-central Bangladesh, where it has reportedly replaced *L. interrogans* as a primary species [71,93–95]. In this study, we identified urinary shedding of the *L. wolffii*-related group in six animals from Ranong and Prachuap Khiri Khan. Similarly, *L. licerasiae* (associated with mild symptomatic infections) was first described in rats and humans in (or returning from) South America [96,97] and has since been detected in Australian swine and leptospirosis-vaccinated dogs in Sri Lanka [48,98]. We identified two animals carrying the *L. licerasiae*-related group in Tak and Ranong. These findings underscore the potential emergence of *L. wolffii*- and *L. licerasiae*-related infections in animal reservoirs in Thailand, as well as across South and Southeast Asia. Although the precise species within the P2 subclade infecting these dogs and cats remains undetermined, the broad geographical distribution of these reservoirs warrants further attention regarding their implications for animal and public health.

This study has several limitations to consider. First, the geographic scope and sample distribution were restricted. Sampling was limited to central, western, and southern Thailand, and sample sizes were unevenly distributed across sites (median = 41.5; range = 12–238). This imbalance may have reduced the statistical power to detect infections in sites with low recruitment, suggesting that these results may not fully represent the epidemiological situation in the northern or eastern regions of the country. Second, the reported prevalence likely underestimates the true infection rate. As our study focused on urinary shedding (leptospiruria) to identify chronic carriers, we did not perform blood sample investigation; consequently, animals in the acute phase of infection (where the pathogen is present in the blood but not yet shed in the urine) were not captured. Furthermore, because *Leptospira* are shed intermittently [25,42], a single-point urine collection may have yielded false-negative results for some carriers. Other factors also contribute to this underestimation, including potential false PCR negatives due to antibiotic administration before urine collection (a factor we could not fully control, though our neutering protocol did not use antibiotics). Third, logistical constraints impacted molecular identification. Due to limited field resources, samples were maintained at 2°C–6°C without pH neutralisation. The resulting exposure to potentially acidic urine, particularly in samples requiring prolonged transport (>24 hours) from distant sites, may have caused DNA degradation and reduced diagnostic sensitivity. Additionally, species identification was limited by low DNA yields and potential primer competition, leading to ambiguous Sanger sequencing reads. While the conserved 443 bp *rrs* gene fragment is a rapid method for identifying major P1 subclade species, it cannot accurately distinguish between members of the P2 subclade due to low genetic diversity in this region [9]. Future studies could achieve higher resolution for P2 species by targeting a more diverse set of markers, such as *lipL32* and *secY* [89], or by employing next-generation deep sequencing.

In conclusion, three classical *Leptospira* species (*L. interrogans*, *L. weilii*, and *L. borgpetersenii*), responsible for human leptospirosis, were predominantly identified in dogs and cats in western and southern Thailand. Additionally, species potentially causing mild leptospirosis (*L. wolffii*- and *L. licerasiae*-related groups) and those with unknown infectivity (*L. yasudae*) were also detected. Our findings highlight the role of dogs and cats as reservoirs of pathogenic *Leptospira* spp., emphasising the need for expanded surveillance, broader-spectrum vaccines to reduce urinary shedding, and public awareness campaigns to mitigate zoonotic risk.

## Supporting information

**S1 File. Identification of *Leptospira* species.**
(DOCX)

**S2 File. Annual case and morbidity rate of study sites reported to the Department of Disease Control, Ministry of Public Health of Thailand, between 2019 and 2022.**
(DOCX)

**S1 Table. Demography characteristics of dogs and cats with urine specimens PCR-positive for pathogenic *Leptospira*.**
(XLSX)

**S2 Table. List of reference *rrs* sequences used for the 16S rRNA sequence analysis.**
(XLSX)

**S3 Table. Frequency of *Leptospira* groups in dogs and cats stratified by study site, age group, sex, and owned status.**
(DOCX)

## Acknowledgments

We are grateful to the staff of the One Health mobile unit, Faculty of Veterinary Science, Mahidol University, and all the volunteers from Soi Dog and the nonprofit organisations for their assistance with sample collection during the neutering process. We would also like to thank all the dog and cat owners who participated in this study.

## Author contributions

**Conceptualization:** Metawee Thongdee, Sarin Suwanpakdee, Janjira Thaipadungpanit.

**Data curation:** Janjira Thaipadungpanit.

**Formal analysis:** Sarin Suwanpakdee, Janjira Thaipadungpanit.

**Funding acquisition:** Metawee Thongdee.

**Investigation:** Metawee Thongdee, Somjit Chaiwattanarungruengpaisan, Weena Paungpin, Sivapong Sungpradit, Sineenard Jiemtaweeboon, Ekasit Tiyanun, Kanin Ruchisereekul.

**Methodology:** Metawee Thongdee, Somjit Chaiwattanarungruengpaisan, Weena Paungpin, Janjira Thaipadungpanit.

**Supervision:** Janjira Thaipadungpanit.

**Validation:** Metawee Thongdee, Janjira Thaipadungpanit.

**Writing – original draft:** Metawee Thongdee, Kanin Ruchisereekul, Janjira Thaipadungpanit.

**Writing – review & editing:** Metawee Thongdee, Somjit Chaiwattanarungruengpaisan, Weena Paungpin, Sivapong Sungpradit, Sineenard Jiemtaweeboon, Ekasit Tiyanun, Sarin Suwanpakdee, Janjira Thaipadungpanit.

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
