## [Decision Letter · Decision Letter 0]

28 Sep 2025

PNTD-D-25-01286Pathogenic Leptospira species identified in dogs and cats during neutering in ThailandPLOS Neglected Tropical Diseases Dear Dr. Thaipadungpanit, Thank you for submitting your manuscript to PLOS Neglected Tropical Diseases. After careful consideration, we feel that it has merit but does not fully meet PLOS Neglected Tropical Diseases's publication criteria as it currently stands. Therefore, we invite you to submit a revised version of the manuscript that addresses the points raised during the review process. Please submit your revised manuscript within 30 days Nov 27 2025 11:59PM. If you will need more time than this to complete your revisions, please reply to this message or contact the journal office at plosntds@plos.org. Please include the following items when submitting your revised manuscript:* A rebuttal letter that responds to each point raised by the editor and reviewer(s). You should upload this letter as a separate file labeled 'Response to Reviewers'. This file does not need to include responses to any formatting updates and technical items listed in the 'Journal Requirements' section below.* A marked-up copy of your manuscript that highlights changes made to the original version. You should upload this as a separate file labeled 'Revised Manuscript with Track Changes'.* An unmarked version of your revised paper without tracked changes. You should upload this as a separate file labeled 'Manuscript'. If you would like to make changes to your financial disclosure, competing interests statement, or data availability statement, please make these updates within the submission form at the time of resubmission. Guidelines for resubmitting your figure files are available below the reviewer comments at the end of this letter. We look forward to receiving your revised manuscript. Kind regards, Brianna R Beechler, Ph.D., DVMAcademic EditorPLOS Neglected Tropical Diseases Stuart BlacksellSection EditorPLOS Neglected Tropical Diseases

Shaden Kamhawi

co-Editor-in-Chief

Paul Brindley

co-Editor-in-Chief

**Additional Editor Comments (if provided):** Two reviewers have several minor suggested revisions on things that should be clarified in the methods or discussed briefly and/or some details that may be better placed in supplement. Please respond to their concerns and queries. **Journal Requirements:**

If the reviewer comments include a recommendation to cite specific previously published works, please review and evaluate these publications to determine whether they are relevant and should be cited. There is no requirement to cite these works unless the editor has indicated otherwise.**Reviewers' comments:** Reviewer's Responses to Questions

**Key Review Criteria Required for Acceptance?**

**Methods**

-Are the objectives of the study clearly articulated with a clear testable hypothesis stated?

-Is the study design appropriate to address the stated objectives?

-Is the population clearly described and appropriate for the hypothesis being tested?

-Is the sample size sufficient to ensure adequate power to address the hypothesis being tested?

-Were correct statistical analysis used to support conclusions?

-Are there concerns about ethical or regulatory requirements being met?

Reviewer #1: The article has the aim to investigate Leptospira infection in dogs and cats in three regions in Thailand and to provide epidemiological data for the purpose of control and prevention of the infection.

The abstract gives the methods used and results of the study. The title recalls the key features of the article and does not add anything more to the findings of the study.

The references are cited correctly and are mostly recent (20 out of 98 are dated before 2015).

The introduction briefly summarizes the current knowledge about the epidemiology of leptospirosis, citing papers reporting data from different countries and referred on studies on dogs and cats.

The introductive part clearly defines the purpose of the research as indicated in the abstract.

The paper presents a cross-sectional study based on samples collected in three different regions in Thailand.

The study design, the criteria for sampling and the methods used are clearly described. The molecular tests are known and supported by references.

The statistical methods are clearly defined.

Reviewer #2: The objectives, experimental design, sampling, and statistical model are compatible with the proposed study. However, a few details regarding the sampling require further information (specifically on the intermittency of bacterial shedding and sample preservation).

Reviewer #3: Yes

Yes

N/A

Yes

Yes

Yes

**Results**

-Does the analysis presented match the analysis plan?

-Are the results clearly and completely presented?

-Are the figures (Tables, Images) of sufficient quality for clarity?

Reviewer #1: The data are presented appropriately: tables and figures are clear with titles, columns and rows labelled correctly and clearly.

The text summarizes briefly the contents of the tables but it is not repetitive.

Reviewer #2: The results are compatible with the proposed hypothesis; however, due to the characteristics of the clinical stage of the disease (the leptospiruria phase), intermittent bacterial shedding in the urine may occur. Therefore, the authors need to, at a minimum, mention that the results may be underestimated.

Reviewer #3: Please ref the reviewer comments attached

**Conclusions**

-Are the conclusions supported by the data presented?

-Are the limitations of analysis clearly described?

-Do the authors discuss how these data can be helpful to advance our understanding of the topic under study?

-Is public health relevance addressed?

Reviewer #1: In the first part of the discussion, a number of papers on the prevalence of Leptospira infection in different countries are cited. The authors are conscious (see lines 327-328) that comparisons between results of different studies are not always possible and that variations depends from a number of factors. Moreover, the authors must be careful with the use of term “prevalence” when they refer to the figures from countries they have considered.

In the following sections the results are discussed and placed correctly into context and are not overinterpreted; the conclusions answer the aims of the study and are supported by results.

The authors are aware of the limit of their research considering that the sample size is not equally distributed across all investigated sites and that only a proportion of positive samples were subjected to species identification: this is due to limitations inherent to the sample quality and the methods adopted.

Reviewer #2: The conclusions are consistent with the stated objectives.

Reviewer #3: Please refer to the reviewer comments attached.

**Editorial and Data Presentation Modifications?**

Reviewer #1: (No Response)

Reviewer #2: No

Reviewer #3: Please refer to the reviewer comments attached.

**Summary and General Comments**

Reviewer #1: The study, despite the limits specified, gives valuable insights suggesting region-specific mitigation strategies to reduce the risk of Leptospira transmission.

Reviewer #2: To the Authors,

First and foremost, I would like to congratulate you on the results obtained, particularly given the substantial sample size. The results are strongly supported by the techniques employed and are, likewise, carefully discussed.

However, a few minor observations and questions need to be clarified in the manuscript:

Positive PCR results from urine samples indicate that the animal is in the leptospiruria phase. However, as bacterial shedding is intermittent during this phase—that is, inconsistent over time—negative results do not allow one to conclude that the patient is not a carrier. This information should be included in the discussion, noting that the results may be underestimated; in other words, the detection frequency could be higher than reported.

Were any of the animals under preoperative treatment (antibiotic therapy)? The administration of antibiotics prior to sample collection may lead to false-negative PCR results.

Was any buffer (e.g., PBS) used to stabilize the urinary pH immediately after collection? DNA extraction from urine can be interfered with by several factors and may even be rendered unviable. As urine pH is acidic, it is important to mix it with PBS immediately after collection to neutralize it.

What was the time elapsed between sample collection and DNA extraction? If DNA is not extracted on the same day, it is recommended that the urine (with stabilized pH) be stored at 5°C and the DNA extracted within the next 24 hours. What procedure was adopted in the study?

I believe these points need to be clarified within the study.

Once again, congratulations on your work.

Reviewer #3: Please refer to the reviewer comments attached.

PLOS authors have the option to publish the peer review history of their article (what does this mean?). If published, this will include your full peer review and any attached files.

Reviewer #1: No

Reviewer #2: No

Reviewer #3: **Yes:** Chandika Damesh Gamage

**Figure resubmission:** While revising your submission, we strongly recommend that you use PLOS’s NAAS tool (https://ngplosjournals.pagemajik.ai/artanalysis) to test your figure files. NAAS can convert your figure files to the TIFF file type and meet basic requirements (such as print size, resolution), or provide you with a report on issues that do not meet our requirements and that NAAS cannot fix.

After uploading your figures to PLOS’s NAAS tool - https://ngplosjournals.pagemajik.ai/artanalysis, NAAS will process the files provided and display the results in the "Uploaded Files" section of the page as the processing is complete. If the uploaded figures meet our requirements (or NAAS is able to fix the files to meet our requirements), the figure will be marked as "fixed" above. If NAAS is unable to fix the files, a red "failed" label will appear above. When NAAS has confirmed that the figure files meet our requirements, please download the file via the download option, and include these NAAS processed figure files when submitting your revised manuscript. **Reproducibility:** To enhance the reproducibility of your results, we recommend that authors of applicable studies deposit laboratory protocols in protocols.io, where a protocol can be assigned its own identifier (DOI) such that it can be cited independently in the future. Additionally, PLOS ONE offers an option to publish peer-reviewed clinical study protocols. Read more information on sharing protocols at https://plos.org/protocols?utm_medium=editorial-email&utm_source=authorletters&utm_campaign=protocols

---

## [Editor Report · Decision Letter 1]

6 Jan 2026

Dear Dr. Thaipadungpanit,

We are pleased to inform you that your manuscript 'Pathogenic Leptospira species identified in dogs and cats during neutering in Thailand' has been provisionally accepted for publication in PLOS Neglected Tropical Diseases.

Best regards,

Brianna R Beechler, Ph.D., DVM

Academic Editor

Elsio Wunder Jr

Section Editor

Shaden Kamhawi

co-Editor-in-Chief

Paul Brindley

co-Editor-in-Chief

---

## [Editor Report · Acceptance letter]

Dear Dr. Thaipadungpanit,

We are delighted to inform you that your manuscript, "Pathogenic Leptospira species identified in dogs and cats during neutering in Thailand," has been formally accepted for publication in PLOS Neglected Tropical Diseases.

Best regards,

Shaden Kamhawi

co-Editor-in-Chief

Paul Brindley

co-Editor-in-Chief
